# Efficacy and Microbiota Modulation Induced by LimpiAL 2.5%, a New Medical Device for the Inverse Psoriasis Treatment

**DOI:** 10.3390/ijms24076339

**Published:** 2023-03-28

**Authors:** Laura Pietrangelo, Annunziata Dattola, Irene Magnifico, Giulio Petronio Petronio, Marco Alfio Cutuli, Noemi Venditti, Antonio Guarnieri, Andreas Wollenberg, Giovanni Pellacani, Roberto Di Marco

**Affiliations:** 1Dipartimento di Medicina e Scienze della Salute “V. Tiberio”, Università degli Studi del Molise, 86100 Campobasso, Italy; 2Dermatology Clinic, Department of Clinical Internal, Anesthesiological and Cardiovascular Sciences, Sapienza University of Rome, 00185 Rome, Italy; 3Department of Dermatology and Allergy, University Hospital, Ludwig-Maximilian University Munich, 80539 Munich, Germany

**Keywords:** inverse, intertriginous, fold, psoriasis, microbiota, *Corynebacterium*, *Staphylococcus*, *Actinobacteria*, *Firmicutes*, treatment

## Abstract

(1) Inverse psoriasis (IP), also known as intertriginous, typically affects the groin, armpits, navel, intergluteal fissure, and external genitalia. Skin lesions are erythematous plaques of inflammatory nature, smooth, well-delimited, non-scaly, and non-infiltrated. Lesions may be accompanied by itching, pain, or burning sensation. The aim of this study is both to investigate the modulation of the skin microbiota induced by IP and, on the other hand, to test the effectiveness of the new biotechnological product LimpiAL 2.5%. (2) Patients affected by IP were recruited in a private practice and treated for 4 weeks with LimpiAL 2.5% exclusively. The clinical effects on the lesion skin were evaluated, and the skin microbiotas before and after treatment were compared. (3) The clinical outcomes reveled a significant beneficial effect of the tested product. At the same time, LimpiAL increased the biological diversity of the skin microbiota and exerted a significant decrease of some *Corynebacterium* species, and the increase of some *Staphylococcus* species. (4) Together, the clinical outcomes and the microbiota analysis suggest that LimpiAL treatment improves the skin condition of affected patients, basically restoring the eubiosis conditions of the affected sites and modulating the bacterial composition of the resident microbiota.

## 1. Introduction

Inverse psoriasis (IP) is a chronic inflammatory skin disease involving the body folds and intertriginous areas, therefore, it is also defined as flexural, intertriginous, or skin-fold psoriasis [1,2]. Axillae, perianal skin, intergluteal cleft, inframammary, genital/inguinal, abdominal, and retro auricular folds are the body areas more frequently affected [2,3]. The generated lesions are somewhat similar to, but less scaly than, plaque psoriasis, which is why the two pathologies are difficult to discriminate, especially when they overlap in the same patient [4]. The only two principal discriminating factors are related to the sites affected and to some peculiar aspects of the skin lesions [5]. More precisely, IP is localized exclusively at sites where two skin surfaces converge and the skin lesions lack, or show less, scaling, are well-demarcated, erythematous, and often with a shiny surface [5,6,7]. The IP symptoms directly result from the involved body area’s anatomy. A moist and warm milieu of fold sites causes the superficial erosions and maceration of tissues, resulting in intense itching, irritation from sweating, and soreness [2,8]. In addition, since the moist and damaged skin provide all the favorite conditions for microbial growth, the development of bacterial and fungal superinfection is also quite frequent, as well as the microbial proliferation at the body fold itself, could promote the IP onset [2,6,9,10,11]. However, the lack of microbiota studies based on advanced sequencing methods makes it hard to establish the role of some microbes detected at affected sites.

Beyond all the identification difficulties, the IP diagnosis is principally clinical and based on the physical examination of skin folds at first, but then of the entire body and mucosae to verify the existence of possible comorbidities and/or concomitant plaque psoriasis [2]. In addition, dermoscopy, skin biopsies with histopathology, and bacterial and fungal cultural methods help diagnose any case of clinical doubt regarding secondary infections, malignancy, or similar diseases [2,12].

The treatment of IP is distinct from plaque psoriasis, and it may be challenging due to the characteristic anatomy of the involved fold areas [2]. The thinner skin of these body sites and the occlusive effect caused by the converging of skin layers often determine side effects hindering the therapy [2,12]. Currently, the Medical Board of the National Psoriasis Foundation recommends low- to mild-potency topical corticosteroids as the first-line therapy [2,3]. Unfortunately, the well-known side effects associated with the prolonged use of steroids make these medicaments applicable only for short-term treatment, i.e., <4 weeks [13,14]. As the alternative first-line therapy for long-term treatment, although with a lower efficacy, topical calcineurin inhibitors (TCI), such as tacrolimus or pimecrolimus, and vitamin D analogues, are recognized by several studies [2,12,15,16]. 

However, at present, the difficulty in IP diagnosis and the lack of specific studies about skin microbiota involvement and other possible therapeutic strategies concur together to generate an important lack of knowledge and, consequently, often inefficient management of the pathology.

This study’s aim was, therefore, to fill at least two of the aforementioned gaps. On the one hand, an in-depth investigation of the microbiota colonizing the lesion areas was conducted using two different advanced approaches of high throughput sequencing to investigate the microbial involvement in the etiopathogenesis of the IP. On the other hand, the new biotechnological medical device, LimpiAL 2.5% (Medical Device, reg. BD/RDM 2112380), was tested as an innovative treatment for inverse psoriasis. LimpiAL 2.5% is a topical preparation specifically formulated for the treatment of non-hyperkeratotic or mildly hyperkeratotic psoriasis, especially for inverse psoriasis. The main component of the device is the patented HAc-40 ingredient (DEPOSIT: WO2022243558A1, Aileens Pharma SRL). The HAc-40 is made by high molecular weight hyaluronic acid conjugated with a bacterial wall fragment from *C. acnes* DSM 28,251 (DEPOSIT: 9517149, Aileens Pharma SRL). The intrinsic chemical–physical characteristics of the HAc-40 strongly support the action of the LimpiAL formula. The high molecular weight of the hyaluronic acid offers the advantage of not being systemically absorbed when topically applied, thus limiting the risk of adverse reactions of systemic origin. Thanks to the conjugation process with the bacterial fragment, HAc-40 acquires better film-forming properties. As a consequence, LimpiAL forms an invisible and non-occlusive film that protects the lesion skin from the external factors, reduces Trans-Epidermal Water Loss (TEWL), and preserves the skin hydration, thus finally promoting the maintenance of skin homeostasis. In addition, previous studies on inflammatory skin diseases successfully treated with products containing HAc-40 demonstrated the ability of these products to re-balance the healthy skin microbiota pattern and restore the eubiosis state [17,18,19]. Furthermore, considering the lesion state of the psoriasis skin, the product has been developed using highly tolerable ingredients and raw materials with high safety profile and purity (Appendix A).

With this aim, we enrolled two cohorts of volunteer patients affected by IP and individuals with healthy skin as CTRL group from a private practice. They were screened for the skin condition at the starting point of the study (time 0) and after 4 weeks (time 1); during that the affected individuals were treated with LimpiAL 2.5%. At same times, skin microbiota samples were collected from all patients as representative of the microbiota before and after treatment. The microbiota samples from the first cohort of patients were analyzed through the NGS Illumina MiSeq approach and the bacterial (16S rRNA) and fungal (ITS 1 and ITS2) microbial components were characterized. The second cohort samples were analyzed through the Shotgun sequencing approach and bacteria, archaea, eukaryote, and virus prevalence was investigated. Both the Illumina amplicons and Shotgun DNA-Seq raw data were analyzed through the Gaia software (Sequentia Biotech sl, Barcelona, Spain). The microbiota of untreated IP patients was compared to the healthy skin one to detect microbial taxa related to the pathology onset. Then, the microbiota of the treated group was compared with the untreated and healthy skin to evaluate how and how much the LimpiAL microbiota differs from the compared ones. All the comparisons were conducted at both quantitative and qualitative levels (alpha and beta diversity evaluation, OTUs interesting variations, DESeq2 analysis). 

Finally, the microbiota results were correlated to the clinical outcomes to evaluate the potential efficacy of the LimpiAL 2.5% as an innovative treatment for inverse psoriasis. 

Together, our results demonstrated the high potentiality of LimpiAL 2.5% as an innovative treatment for inverse psoriasis. LimpiAL induced an evident amelioration of the clinical skin conditions and significantly modulated some microbial species that consequently could be suggested as directly involved in the pathogenic process or in restoration of the homeostatic skin conditions. Overall, interesting modulation trends induced by LimpiAL resulted for the *Actinobacteria* and *Firmicutes* phyla, and especially for the *Corynebacterium* and *Staphylococcus* genera.

## 2. Results

### 2.1. Patients Recruited

At the first stage of analysis, 24 patients were enrolled. Fourteen patients with healthy skin and without IP lesions were included in the CTRL group. Conversely, the other 10 patients presented IP lesions in different body areas. Only three agreed to the LimpiAL treatment and took part in the T1 group, whereas the other seven patients were included in the T0 lesion but not treated group.

At the second stage of analysis, a total of 14 patients were enrolled. Four showed IP at the mammary fold, three others at the inguinal fold, one at the axillary hallow, one at the elbow, two at the knee, one at the popliteal cavity, one at the abdominal fold, and one at the feet hollow. They all accepted LimpiAL cream as the unique treatment for the subsequent 4 weeks. Thus, all of them were evaluated for clinical outcomes as well. On the contrary, only seven patients agreed to the microbiota sampling from the affected areas. They were the three patients with IP at the mammary fold, two at the inguinal fold, one at the axillary hallow, and the other one at the elbow. The latter one (IP at elbows) did not come back after treatment so was sampled at time 0 only (T.0). In summary, all 14 patients were evaluated for the clinical outcomes, for seven of them, the microbiota before treatment (T.0) was investigated and for five of them, the microbiota after LimpiAL treatment (T.1) was also investigated.

### 2.2. Clinical Outcomes 

In the first stage of the study, only three patients agreed to use LimpiAL cream as a unique treatment for the prescribed time. Despite the low number of observed cases and the high inhomogeneity of the sampled areas, a general appreciable amelioration of the lesion condition was observed for all treated patients (almost 60%). Due to the aforementioned characteristics of samples, unfortunately, not many other speculations can be made about the effectiveness of LimpiAL treatment at this analytic step.

Conversely, a more detailed evaluation of the clinical outcomes has been achievable at the second analysis stage. Among the second cohort of patients, the skin condition was significantly ameliorated by evaluating the skin parameters scores. As indicated in Figure 1, for only one of the 14 treated individuals, the treatment failed, whereas for the other 13, the evaluated skin parameters highlighted a significant clinical benefit of the LimpiAL preparation. 

Even considering the failure case, overall, the PASI score varied from the mean value of 6 ± 3 SD to 3 ± 3 SD, with a mean decrease of 64% (Figure 2a). The DLQI index showed an even higher variation, from the initial mean value of 14 ± 6 SD to 3 ± 5 SD, and incurring a percentage reduction of 77% (Figure 2b). Finally, the itch VAS score decreased from 38 ± 21 SD to 8 ± 18 SD, with a percentage reduction of 85% (Figure 2c). 

### 2.3. DNA Extraction

The extraction of genetic material from the swabs produced a poor but homogeneous yield of DNA. An average amount of 0.60 ng/mL was obtained for CTRL samples, 0.66 ng/mL for T0 and T.0 samples, and 0.75 ng/mL for T1 and T.1 samples. Therefore, an average quantity of DNA of 0.70 ng/mL was obtained overall from all collected samples. 

### 2.4. Analysis of the Microbiota Modulation 

#### 2.4.1. First Analysis Step: The Amplicons Sequencing Analysis of Bacterial and Fungal Microbiota

The sequencing of the 16S rRNA and ITS amplicons produced 915,161 paired-end reads. The trimming process retained 780,155 high-quality reads for the downstream analysis (85%) through quality filtering and the chimeric reads removal. Of these, a total of 504,954 reads were detected for the CTRL group, with values per sample ranging between the interval 19,197–73,712 and a mean value of 36,068 reads (Figure 3a). For the T0 group, 183,554 reads were obtained. Values for samples ranged between 9987 and 39,736 reads and a mean value of 26,222 (Figure 3a). Finally, the sequencing of the T1 group produced in total of 91,647 reads, in the interval between 26,025 and 36,188 reads with a mean value of 30,549 (Figure 3a).

The retained sequences matched with a total of 13,506 operational taxonomic units (OTUs) of the Amplicon 16S and Amplicon ITS1 and ITS2 databases. In detail, 9014 OTUs were detected for the CTRL group, 3267 for T0, and 1225 for the T1 group. The CTRL samples accounted for a mean of 644 OTUs and the T0 and T1 ones for 467 and 408 OTUs, on average (Figure 3b). 

No significant difference was found between the reads number and the detected OTUs between the CTRL, T0, and T1 groups (Figure 3). The statistical comparison between groups was performed through the Kolmogorov–Smirnov test for not normally distributed series through the ANOVA analysis for normal distributions of values (Appendix A).

Similarly, no significant difference was found between the alpha diversity indexes since the Kolmogorov–Smirnov test, and Welch’s *t*-test produced a *p*-value > 0.05 for all comparisons (Appendix A). For the beta diversity, the PCoA of Bray–Curtis distances also did not show any differential clustering of T1 samples due to LimpiAL treatment (Appendix A). 

In spite of this, the bacterial and fungal microbiotas of all groups were compared both at quantitative and qualitative levels. The comparisons T0 vs. CTRL, T1 vs. CTRL, and T1 vs. T0 were conducted.

As shown in Figure 4, the highest number of modulated OTUs can be detected by comparing the T0 microbiota (affected, not treated patients) to the healthy microbiota (CTRL) (Figure 4a, orange). 

This finding confirms that, as conceivable, the pathology induces a consistent modulation of OTUs in the skin microbiota. On the contrary, comparing the T1 microbiota to the healthy one, a significantly lower number of OTUs result in modulating (Wilcoxon test, *p* = 0.0313) (Figure 4a, blue vs. orange). This suggests that the microbiota induced by treatment, the T1 microbiota, is more similar to the healthy one than the untreated microbiota, T0. Nevertheless, the number of OTUs that distinguishes the T1 microbiota from the T0 microbiota is significantly lower than that which distinguishes the T1 from the healthy microbiota (Wilcoxon test, *p* = 0.0313) (Figure 4a, violet vs. blue). These results suggest that, even if the T1 microbiota is less diverse from the healthy one, it is more similar to the T0 microbiota. Then, going deeper into the OTUs classification, under the LimpiAL treatment, although on one side, a significantly lower number of OTUs (Wilcoxon test, *p* = 0.0313) are negatively and positively modulated (Figure 4c,d blue vs. orange), on the other side, a significantly higher number of exclusive OTUs (Wilcoxon test, *p* = 0.0313) are detectable (Figure 4e, blue vs. orange). Simultaneously, a lower number of exclusive OTUs (Wilcoxon test, *p* = 0.0313) differentiates the T1 microbiota from the T0 (Figure 4e, blue vs. violet). These latter OTUs could be responsible for the microbial community routing to the healthy microbial set. Overall, in both comparisons, with respect to the CTRL and to the T0 microbiota, the LimpiAL treatment revealed a clear trend of implementing the microbial diversity. Finally, no significant variation regarding the differential OTUs resulted.

Regarding the percentage amount of the under, over, and exclusive OTUs in each compared microbiota, the OTUs modulated in T0 microbiota with respect to the CTRL, thus by the pathology itself, are principally over-modulated microbial phyla, classes, and order. In contrast, the exclusive OTUs are attributable to families, genera, and species taxa (Figure 5). 

On the contrary, the OTUs modulated by the treatment with respect to the CTRL are mainly exclusive OTUs detected incrementally from the phylum up to the species level. The incremental rate of exclusive OTUs may suggest that LimpiAL acts on metabolic pathways common to more phyla that diversify incrementally up to the species pool.

Going ahead to the qualitative analysis of compared microbiota, the DeSeq2 differential analysis identified a few differentially detected OTUs (Figure 6). 

In detail, in the T0 microbiota, the class of *Cytophagia* was significantly decreased by 3.06 logFC (Figure 6, Appendix A). 

For the T1 microbiota, a significant increase of at least 2 logFC was verified for the *Firmicutes* phylum, the class of *Bacilli*, and orders of *Bacillales*, *Mycoplasmatales*, and *Actinomycetales* (Figure 6, Appendix A). Finally, the *Cyanobacteria* phylum significantly increased in the T1 microbiota compared with the T0 microbiota (Figure 6, Appendix A).

Regardless to the statistical significance of variations, other interesting modulation trends were highlighted (Figure 7). 

Comparing the ten most under and ten most over-modulated OTUs of T0 and T1 microbiotas (with CTRL as reference) between each other, it resulted that under LimpiAL treatment, *Corynebacterium afermentans* was more reduced by 1 logFC. In contrast, *Lactobacillus iners* was less reduced by more than 4 logFC (Figure 7). In addition, under treatment, the increment verified in the T0 condition for *Dermabacter hominis* was braked, and the prevalence of this species was lower by about 3 logFC (Figure 7). In other words, it is thinkable that LimpiAL acts against some species of *Corynebacterium* and *Dermabacter* genus, but preserves some *Lactobacillus* species. Moreover, only for T1 microbiota, three different *Staphylococcus* species were detected among the most over-modulated OTUs, i.e., *S. gallinarum*, *Staphylococcus* sp. *B-3*, and *Staphylococcus* sp. *SBT120*. This latter observation suggests that LimpiAL probably stimulates the biological diversification of the *Staphylococcus* genus (Figure 7).

#### 2.4.2. Second Analysis Step: Shotgun Sequences Analysis of Untreated and LimpiAL-Treated Microbiota of All Body Areas 

The Shotgun sequencing produced 259,390,896 paired-end reads. The subsequent trimming, quality filtering, and removal of chimeric reads retained 240,330,709 high-quality reads for downstream analysis (93%). A total of 155,842,815 were detected for T.0 group and 84,487,894 for T.1 group. Among T.0 samples, the detected reads ranged between 14,185,462–32,446,627 with a mean value of 22,263,259, whereas between the range 4,981,497–26,723,198 with a mean value of 16,897,579 for T.1 samples (Figure 8a).

The subsequent matching with the Gaia WGS and WTS databases for prokaryotes, fungi, and viruses identified a total of 123,873,227 OTUs for the T.0 group with a mean value per sample of 17,696,175, and 65,211,197 for the T.1 group with a mean value per sample of 13,042,239 matchings (Figure 8b).

Although there was a different numerosity of the two sample groups, no significant difference was found for the number of reads (unpaired Student’s *t*-test, *p* = 0.3024) or for the number of corresponding OTUs (unpaired Student’s *t*-test, *p* = 0.3099). 

The alpha and beta diversity of untreated (T.0) and LimpiAL treated (T.1) microbiotas were evaluated at all taxonomic levels. No significant difference was found for the alpha diversity indexes among samples (Table 1).

Similarly, for the beta diversity, the PCoA of calculated Bray–Curtis distances did not show any differential clustering of T.1 samples as an effect of LimpiAL treatment (Appendix A). 

Then, going deep into the qualitative composition of the two microbiotas, at the domain level, the T.0 samples, thus the skin microbiota of patients affected by IP, was dominated by the bacteria component (98.5%). In contrast, only the remaining 1.5% was shared between viruses (1%), eukaryota, and archaea (<1%, together) (Figure 9). 

The statistical analysis confirmed that no significant difference was appreciable between the domain abundances of T.0 and T.1 microbiotas (unpaired Student’s *t*-test for normally distributed series and Kolmogorov–Smirnov test for not normally distributed, *p* > 0.05) (Figure 10). 

At the phylum level, a high prevalence of *Proteobacteria* was detected in both groups, with a calculated frequency of 95% for T.0 and 94% for T.1 microbiota (Table 2). 

At lower taxonomic levels, the prevalent *Proteobacteria* portion was deeply characterized as classes of *Betaproteobacteria* and *Gammaproteobacteria*, with similar frequency in both groups, 78% and 11–12%, respectively, and *Alphaproteobacteria* at a 3% frequency in both T.0 and T.1 microbiotas (Table 2). At the genus level among the *Betaproteobacteria*, the genera *Cupriavidus* and *Ralstonia* accounted for the higher abundances, the first at 66% and 65% in T.0 and T.1 groups, respectively, and the latter at 9% in both groups (Table 2). Among *Gammaproteobacteria*, the genus *Pseudomonas* accounted for 11 and 10% of the prevalence (Table 2). Finally, *Cupriavidus metallidurans* was identified as the most abundant in both T.0 and T.1 microbiotas, with frequencies of 65% and 64% (Table 2). Furthermore, *Ralstonia pickettii* represented 7% in both groups, and *Pseudomonas aeruginosa* and *Pseudomonas fluorescens* had a frequency in both groups of 8% and 2%, respectively. The genera and species belonging to the *Alphaproteobacteria* resulted at much lower frequencies, therefore, they were not included among the most representative ones. In addition, the statical comparison between the distributions of the percentage at the upper (phylum and class) and lower (genus and species) taxonomic levels confirmed that the same taxa result was similarly represented in both T.0 and T.1 microbiotas (paired Student’s *t*-test for normal distributed series and Wilcoxon test for not normal distributed ones, *p* > 0.05) (Figure 11).

Consistently, the DEseq2 analysis did not find any significant variation between T.0 and T.1 microbiotas.

#### 2.4.3. Second Analysis Step: Shotgun Sequence Analysis of Untreated and LimpiAL-Treated Microbiota Concerning the Body Area 

Since there was a low number of samples per body area, no statistical comparison between the T.0 and T.1 microbiotas was possible regarding the number of reads and OTUs, the frequencies of principal biological domains, and alpha and beta diversity indexes. Conversely, the DESeq2 analysis was achievable, at least for the mammary and inguinal areas.

Only the species *Gardnerella vaginalis* was detected as differentially modulated for samples from the inguinal fold (Figure 12). 

Since the typical niche of this germ is the vaginal microbiota and its recovery on the skin surface is quite unusual, this finding is certainly explained by a microbe translocation process from other sites, rather than an in-situ enrichment due to the IP. 

More interesting results were obtained for the samples from the mammary fold. The LimpiAL treatment showed a significant inhibitory action with respect to certain components belonging to the *Actinobacteria* phylum, i.e., the two species *Corynebacterium striatum* and *Corynebacterium jeikeium,* which decreased by 10 and 7 logFC, respectively (Figure 12). A similar inhibitory effect resulted against microbes belonging to the *Firmicutes* phylum; the *Finegoldia* genus and the *Streptococcus agalactiae* were both significantly decreased by 8 logFC (Figure 12). Among the same phylum of *Firmicutes*, LimpiAL stimulates the growth of *Staphylococcus pasteuri*, which increased by 8 logFC (Figure 12).

## 3. Discussion

This study was conceived with two distinct steps of analysis conducted with different patient cohorts and using diverse analytic approaches. In both analytic stages, patients affected by IP were subjected to the treatment with LimpiAL 2.5%, a new topical preparation conceived for the treatment of psoriasis and inverse psoriasis lesions. The main component of this device, the patented HAc-40 ingredient (DEPOSIT: WO2022243558A1, Aileens Pharma SRL), is made by high molecular weight hyaluronic acid (HA) conjugated to the bacterial cell wall (c-40) of *C. acnes* DSM 28,251 (DEPOSIT: 9517149, Aileens Pharma SRL). Previous studies demonstrated that other products containing the HAc-40 ingredient were successfully used to treat inflammatory skin diseases and suggested the potential role of HAc-40 as an innovative skin microbiota modulator [17,18,19]. Therefore, in the present study, we tested the potential action of the LimpiAL topical cream containing the 2.5% of HAc-40 (dried % concentration, Appendix A) on the lesion and microbiota dysbiosis induced by the inverse psoriasis disease. 

At the first analysis stage, three patient groups were enrolled, with each group made up of different individuals (Figure 13). The CTRL group was composed of 14 patients not affected by inverse psoriasis, the T0 group of seven patients affected at different body sites, and the T1 group comprised three patients affected by IP and who agreed to undergo the LimpiAL 2.5% treatment. The skin microbiota was sampled and characterized for all participants. In contrast, the low number of patients treated with LimpiAL at this step allowed only a summative evaluation of the clinical outcomes. 

Instead, two patient groups were set out at the second stage of analysis, and these two groups were made up of the same affected individuals who were sampled at different times (Figure 13). In detail, 14 patients affected by IP were enrolled in total. All of them accepted to apply LimpiAL as a unique treatment for the prescribed time, whereas only seven agreed to the microbiota sampling. Therefore, all patients were evaluated for the clinical outcomes, whereas only seven took part in the sampling groups for the microbiota analysis. More precisely, the samples obtained from these seven patients before treatment were considered as T.0 samples, and those collected after the treatment as T.1 samples. Interestingly, in this case, the same body area of each patient was sampled at both times, and then the microbiota modulation was also linkable to the body sites sampled. 

For both analyses, the microbiota swabs were collected by the same operator and then managed from the same lab for genomic extraction and sequencing. Indeed, although at a low concentration, the high homogeneity of the DNA yield confirmed the adequate standardization of sampling procedures and genomic extraction. 

In addition, two different sequencing approaches were then run. In the first analytical step, the NGS Illumina sequencing was implemented to profile only the bacterial (16S rRNA gene) and fungal (ITS genes) microbiotas from healthy (CTRL), lesion (T0), and treated lesion skin (T1). The second analytical step was instead based on the Shotgun sequencing approach to deepen, at all domain levels, the differences between the before (T.0) and after the LimpiAL treatment (T.1) microbiota. 

Regardless of the sequencing approach applied, the microbiota raw data were managed through the Gaia software (Sequentia Biotech sl). At both analytic levels, the microbiota of untreated IP patients was compared to the microbiota of healthy individuals and all the variations induced by the onset of IP disease were highlighted. Furthermore, the microbiota of patients treated with LimpiAL was compared with the untreated and healthy ones to point out the interesting microbial variations induced specifically by the product application on the lesion and healthy skin. All comparisons were performed at both quantitative and qualitative levels, evaluating the alpha and beta diversity of the microbial communities identified and all the significant (DESeq2 analysis) and interesting variation trends of the detected OTUs. Finally, the results concerning the microbiota were correlated to the clinical data to understand the potential action of LimpiAL 2.5% both on the skin microbiota and on the clinical condition of the IP lesions.

Despite the different sequencing methods, both for the NGS and Shotgun sequencing, a similar total number of reads and OTUs was obtained between the considered groups of each analysis (Figure 3 and Figure 8). This finding, on the one hand, again confirms the standardization of the microbiota recovery from swabs. On the other hand, it demonstrates that regardless of the sequencing method applied, the total abundance of the microbiota is similar between the healthy, IP affected, and affected and treated skin. As a consequence, we can conclude that if LimpiAL induces a modulation, it could not be related to the quantitative variation of the total abundance of microbes. The only study we found in the literature about the IP microbiota characterization was based on NGS sequencing of only bacterial libraries, making it difficult to compare with our results. However, the mean value of reads and the number of OTUs we obtained was consistently higher considering it includes the fungal amplicons, i.e., our average number of reads of 38,132 ± 15,734 and a total of 13,506 OTUs vs. 21,787 ± 10,513 average of reads and 1517 OTUs of the reference study [11].

The alpha and beta diversity indexes were then evaluated at all taxonomic levels to verify the diversity within samples and between groups. On both analytic steps, the statistical comparison of the alpha diversity indexes revealed congruently that the microbiota of healthy skin, untreated, and treated lesions were characterized overall by a similar richness (OTUs number and Chao1 indexes) and evenness (Shannon index) (Table 1 and Appendix A). At the same time, for the beta diversity, the PCoA analysis of Bray–Curtis distances at all taxonomic levels did not highlight any differential clustering of samples induced by the pathology or treatment factor (Appendix A). Even if a literature reference about IP is not available, these results are not surprising compared to the finding about the more investigated plaque psoriasis. Although numerous studies evaluated the alpha and beta diversity of the psoriasis lesion microbiota, the lack of standardized protocols, different numbers of samples, sampling procedures, sites sampled, and sequencing and analysis methods caused the obtainment of contrasting results [20]. Therefore, whereas some studies demonstrated an increase in alpha diversity, others suggested a decreasing trend [20,21,22].

Similarly, all studies found inconsistent results for beta diversity [20,21,22,23]. This suggests that, due to the many factors influencing the alpha and beta diversity determinations, it is difficult to compare the results of different studies since they are frequently inconsistent. Then, the unexpected congruence of our results between the different analysis stages, sequencing methods, and distinct patient cohorts support our findings’ validity. However, we cannot ignore that the inhomogeneity of sampling sites, the different number of patients for each group and analytic steps, as well as the short-term treatment, can hide significant variations of the overall alpha and beta diversity. 

Going further into the microbial diversity of microbiotas and deepening the characterization of the modulated components, interestingly, the untreated lesion showed a significantly higher number of modulated OTUs than the LimpiAL treated lesion (Figure 4a, orange vs. blue). Therefore, LimpiAL treatment conditions the IP microbiota, leading it nearer to the healthy microbial composition. Congruently, after LimpiAL treatment, a significantly lower number of OTUs were under- and over-modulated (Figure 4c and Figure 5d, blue vs. orange). On the contrary, LimpiAL increased the amount of some exclusive OTUs, i.e., those OTUs specifically and uniquely detected in treated samples, suggesting a clear action of increasing the microbial diversification of the IP microbiota rather than an over- or under-modulation of such microbial components (Figure 4e, blue vs. orange).

Since previous studies have demonstrated that non-lesion microbiota presents a higher diversity than the lesion microbiota for IP, we are even more confident to confirm what we already verified by the modulated OTUs, i.e., the LimpiAL treatment tends to restore the healthy microbial composition [11,24]. Further confirmation of our hypothesis comes from the previous study about LimpiAD foam, a diverse formulation of the same product that, although applied as a preventive strategy against the development of pressure ulcers, revealed the same tendency to revert the microbiota to the healthy composition [18]. What about the taxonomic classification of the modulated OTUs? The major percentage of the modulated OTUs of untreated microbiota were identifiable as enriched microbial phyla, classes, and orders (40–43%), whereas the exclusive ones were more abundant at family, genus, and species levels (39–71%) (Figure 5). On the contrary, the prevalence of the exclusive OTUs induced by LimpiAL is highlighted at all taxonomic levels, incrementally from the upper to the lower ranks (Figure 5). This lets us speculate that the pathology onset stimulates the increment of some species, classes, and orders already part of the skin microbiota, and the occurrence of new probably pathology-dependent families, genus, and species. On the other hand, LimpiAL counteracts the over-modulation induced by the pathology, increasing the microbial diversification, probably acting on molecular pathways common to diverse microbial phyla, given the diversification interests incrementally all the other lower ranks. 

Through the DEseq2 analysis, the OTUs differentially detected in the studied microbiotas were also identified. From the first analysis step, we verified that the class *Cytophagia* (phylum of Bacteroidetes) was significantly decreased in the untreated lesion microbiota (−3.1 logFC). In contrast, the *Firmicutes* phylum, class of *Bacilli*, and orders of *Bacillales Mycoplasmatales* and *Actinomycetale* were all significantly enriched by at least 2.6 logFC (Figure 6). Ultimately, the DESeq2 analysis also demonstrated that LimpiAL induced specific modulations of the IP microbiota that differ from those induced by the pathology itself. 

Regardless of the statistical significance of the variation, other noteworthy modulation trends at the species level were found. Interestingly, in the LimpiAL induced microbiota, *Corynebacterium afermentans* was more reduced by 1 logFC, whereas *Lactobacillus iners* was reduced by more than 4 logFC, compared to the results of the lesion untreated microbiota (Figure 7). Conversely, the *Dermabacter hominis* prevalence was reduced by about 3 logFC (Figure 7). In addition, the treatment stimulated three different *Staphylococcus* species exclusively. Together, these data suggested that LimpiAL likely exerts an inhibitory effect against the genera *Corynebacterium* and *Dermabacter*, restores the *Lactobacillus* genus, and increases the microbial diversification among the *Staphylococcus* genus. 

The Shotgun analysis was performed regardless of the body area sampled, which did not highlight any significant difference between the microbial communities before and after treatment. A similar composition at the domain level with a consistent prevalence of the bacteria (at least 97%) in both microbiotas was found (Figure 9 and Figure 10). In addition, at the lower taxonomic level, the two microbiotas did not differ significantly (Figure 11). Both showed a marked prevalence of the *Proteobacteria* phylum, at least 90%, of the class of *Betaproteobacteria*, at least 75%, of the genus *Cupriavidus,* 61% at least, and of the species *Cupriavidus metallidurans,* at least 60% (Table 2). 

The DESeq2 analysis conducted with regard to the body areas revealed that LimpiAL exerts its effect principally on the *Actinobacteria* and *Firmicutes* phyla (Figure 12). Among *Actinobacteria*, two *Corynebacterium* species were significantly and strongly inhibited, i.e., *Corynebacterium striatum* and *Corynebacterium jeikeium*, with substantial decreases of 10 and 7 logFC, respectively (Figure 12). Among *Firmicutes*, the *Finegoldia* genus and the *Streptococcus agalactiae* species were both similarly significantly decreased by 8 logFC, whereas the *Staphylococcus pasteuri* increased by 8 logFC (Figure 12).

Since there is a lack of IP microbiota referring studies and due to the complexity of our study setting, we can discuss these latter results, highlighting preferentially those findings that are congruent between the two analytic steps. 

Overall, our results suggested that LimpiAL exerts an inhibitory action against some *Corynebacterium* species. *Corynebacterium striatum* and *Corynebacterium jeikeium* were identified as significantly reduced by the Shotgun analysis, as well as *Corynebacterium afermentans* being identified among the interesting reduction trends, although not significant, by the NGS analysis. The involvement of these species in the IP onset is realistic since the *Corynebacterium* genus has already been identified as a prevalent colonizer of the moist body areas; as the microbial determinant of erythrasma, a cutaneous disease that, similarly to the IP, affects the intertriginous areas; and as directly correlated with the severity of lesions in common psoriasis [9,11,25]. Another interesting effect of the treatment was detected for the *Staphylococcus* genus at both analysis levels. The species *Staphylococcus pasteuri* was significantly increased by the treatment, and the species *Staphylococcus gallinarum*, *Staphylococcus* sp. *B-3*, and *Staphylococcus* sp. *SBT120* showed incremental trends in the first analysis step, exclusively induced by LimpiAL. 

As a reference study reports, the *Staphylococcus* genus is a common colonizer of moist areas such as the body folds. A particularly high prevalence of *S. lugdunensis*, *S. hominis,* and *S. epidermidis* has been found at axillary, abdominal, mammary, and inguinal folds, and the superior intergluteal cleft [11]. At the same time, the involvement of this genus and, in particular, of the species *Staphylococcus aureus* and *Streptococcus pyogenes,* in the pathologic mechanisms of common psoriasis has been highlighted by several studies [17,26,27,28]. However, since the species we identified as enriched do not match those identified at fold sites or related to the pathologic process that affects the skin, we can deduce that LimpiAL increases the diversification among this genus by promoting some beneficial competitors of the pathologic species. This potential mechanism is further supported by the literature findings about the species we found as significantly increased by LimpiAL, i.e., *Staphylococcus pasteuri*. This species produces pasteuricin, a novel bacteriocin with strong antimicrobial activity against staphylococci, including methicillin-resistant *S. aureus* (MRSA) and gram-positive bacteria [29,30]. Since testing of the antimicrobial efficacy, the use of pasteuricin as an alternative antimicrobial treatment against *S. aureus* infections, comprising skin infections, has already been suggested [31]. We can then conclude that by promoting an increase in this species, LimpiAL induces a biological regulation mechanism based on the intra-genus, but also potentially intra-species, competition. 

In the end, regarding the overall clinical outcomes, an appreciable amelioration of the IP lesions was observed after the LimpiAL treatment. Our patients’ PASI score was reduced by an average of 64%, the DLQI score by 77%, and the itch VAS score decreased by 85% (Figure 1 and Figure 2). Thus, we can conclude that the LimpiAL 2.5% treatment significantly improves the clinical conditions of the IP lesion within just 4 weeks of treatment, and its beneficial effect is comparable to that of the first line therapy [2,3,12,15].

## 4. Materials and Methods

### 4.1. Patients Recruitment Procedure

This study was conceived as a single-center study and was conducted in two separate investigation steps. All procedures regarding patient recruitment and sampling were conducted by the same specialized dermatologist in a private practice in Rome (central Italy), in compliance with ethical rules for human experimentation (see Declaration of Helsinki). After the adequate information disclosure by clinicians, all the recruited patients provided their informed consent freely.

In the first investigation step, three different conditions were considered among the study population: patients without any psoriatic skin lesions (control group, CTRL), patients with psoriasis lesions (untreated group, T0), and patients with skin lesions willing to be treated with the LimpiAL formulation (treated group, T1) (Figure 13). 

Patients screened and evaluated by dermatologists were enrolled in February 2021. The inclusion criteria for T0 and T1 groups were the age of at least 18 years, an established diagnosis of psoriasis for at least 6 months, a Psoriasis Area and Severity Index (PASI) ≥ 3, a Dermatology Life Quality Index (DLQI) ≥ 3, and a Visual Analogue Scale (VAS) score ≥ 4 for pain intensity. In addition, for the T1 group, the acceptance to use LimpiAL as a unique treatment for all the study duration was ensured.

The inclusion criteria for the CTRL group were only the age of at least 18 years and the absence of any psoriasis lesions or symptoms. 

The exclusion criteria were the application of cortisone products during the 3 weeks prior to enrollment, history of malignancy (non-melanoma skin cancers excluded), chronic or recurrent infectious diseases, ongoing infectious diseases (HCV, HIV, HBV, TBC, SARS-COV), pregnancy and breastfeeding, systemic pathologies that could influence the patient’s health and safety, and interfere with the skin reaction to treatment.

In the second investigation step, the study was deepened and focused on comparing the affected patients before and after LimpiAL treatment. According to the same inclusion and exclusion criteria, new patients’ groups were enrolled in January 2022. Different from the first step of the analysis, the same patients composed the two before and after treatment groups. Then, the same body areas were sampled before and after treatment (Figure 13). To highlight this diversification, the two groups in the second analysis step were named as T.0 and T.1, underling the two-time points of the same area sampling (Figure 13). 

In summary, in the first step, the CTRL, T0, and T1 groups were made up of patients unrelated to each other, considered as representative of their skin condition, i.e., healthy, lesioned, lesioned and treated skin. In the second step, T.0 and T.1 groups were composed by the same patients sampled at two different time points, before and after treatment. In such a way, in the first run, we investigated the pathology determinants and how LimpiAL affects overall skin condition, whereas in the second run, we deepened the study on the affected patients at the individual level, considering the evolution of the pathology under the LimpiAL treatment.

### 4.2. Treatment and Sampling

The body area affected by IP was detected, and a 5  cm^2^ area of skin was chosen for the microbiota sampling. For the CTRL patients, almost the same body areas of the T0 and T1 groups were chosen for the sampling of healthy skin (not-lesion skin) microbiota. 

Regardless to the intrinsic characteristics of the treated groups between the two analytic steps, the T1 and T.1 patients were sampled after they treated the psoriasis lesion with LimpiAL, as prescribed. In detail, they applied LimpiAL for 4 weeks every day, twice a day, and after accurate water-washing of the sampling site.

The microbiota sampling was conducted by the dermatologist in charge using the e-nat swabs (Copan eNat^®^, COPAN ITALIA spa, Brescia, Italy), as recommended by the manufacturer’s procedure, in sterile conditions, applying a firm pressure, rubbing the swab back and forth vigorously 50 times (for 30 s), in a parallel direction to the skin surface. The microbiota samples were transported to the lab at a controlled temperature of 4 °C and stored at −80 °C until further processing.

Only in the second stage of the study, since the attention was directed to the effect of LimpiAL treatment on psoriatic symptom evolution, photos of the sampled areas were also collected at both times points T.0 and T.1, as well as the skin parameters, were registered (Figure 13).

### 4.3. Clinical Evaluation

The overall variation of the clinical skin conditions after LimpiAL treatment was evaluated for both patient cohorts. However, a more detailed investigation about the clinical outcomes was only achievable for the second analysis stage, when both the skin parameters were registered, and lesions photos were captured. The PASI, DLQI, and itch VAS scores before and after treatment were statistically compared through the non-parametric Wilcoxon test (*p* ≤ 0.05). In addition, the captured photos of the lesion were used to by-eye evaluate the LimpiAL effect on the IP lesions.

### 4.4. Molecular Analysis

The sampled swabs were processed at Biolab (Ascoli Piceno, Italy). The study setup is briefly described below, and a schematic is shown in Figure 13. Firstly, the genomic DNA extraction was performed using the ChargeSwitch™ gDNA Normalized Buccal Cell Kit (Invitrogen™). The purified DNA from CTRL, T0, and T1 samples (first investigation step) was subjected at the same time to the amplification of the bacterial 16S rRNA and fungal ITS marker genes, generating amplicons libraries of both bacterial and fungal microbiota. The amplicons were sequenced through the 2 × 300 bp NGS Illumina MiSeq system (100 K reads), profiling the microbiotas associated with the CTRL, T0, and T1 conditions. 

Conversely, the DNA isolated from the T.0 and T.1 swabs was sequenced through the Shotgun DNA-Seq approach (20 M reads).

### 4.5. Data Analysis

The raw data from Illumina amplicons and Shotgun DNA-Seq were analyzed through the Gaia software (Sequentia Biotech sl). Briefly, forward and reverse FASTQ files were decompressed, evaluated for the sequencing quality (Quality Control, QC), and trimmed using the default setting of the GAIA algorithm. The trimmed reads were merged into overlapping paired-end reads, and the high-quality reads only were retained. 

For the Illumina amplicons, the bacterial libraries were matched against the database Amplicon 16S (release 2020), and the fungal libraries against the database Amplicon ITS1 and ITS2 (release 2020).

The full-length Shotgun genomes were simultaneously matched against the databases WGS and WTS Prokaryotes (bacteria and archaea), WGS and WTS Fungi, and WGS and WTS Viruses (all released on 2020).

Gaia Software from NCBI genomes sequences custom to the reference databases. The OTUs given by Gaia are based on at least 97% identity working at the species level.

The OUTs table was matched with the taxonomy entries to generate a taxonomy table reporting the abundance of all OUTs in each microbiota. The taxonomy table profiled the microbiota of all samples and was used to conduct the statistical analysis. 

Due to the different intrinsic nature of our data (amplicons libraries and full-length genomes), a diverse and specific analysis process was implemented for the two sets of samples. 

### 4.6. 16S and ITS Libraries Analysis

The microbiota of untreated psoriasis patients was compared with the microbiota of healthy skin (T0 vs. CTRL) to detect microbiota components significantly influenced by the pathology onset. Then, the microbiota of the treated group T1 was compared both with the T0 untreated group and with the CTRL group to evaluate the significant variations induced by the treatment in the lesion microbiota (T1 vs. T0), and how much the LimpiAL treated microbiota diverges from the healthy skin microbiota (T1 vs. CTRL). All the comparisons were conducted at both quantitative and qualitative levels. 

At the quantitative level, the number of OTUs detected as modulated, differential, and under, over, and exclusively represented was evaluated and compared between all the interesting comparisons. The normality of data distributions was assessed through the Shapiro–Wilk test (α = 0.005), and then the significance of comparisons was tested through the non-parametric Wilcoxon test (*p* ≤ 0.05).

The percentage distribution of differential, over, under, and exclusive OTUs of all interesting comparisons (T0 vs. CTRL, T1 vs. CTRL, and T1 vs. T0) was calculated along all taxonomic levels. The more represented category of OTUs among the ones listed before (higher percentage) identified the prevalent type of modulation induced and the taxonomic level it acted predominantly on. 

Instead, for the qualitative evaluation before all, the DESeq2 analysis was performed to identify the OTUs differentially represented in the comparisons of interest along all the taxonomic levels. 

Finally, the investigation was deepened to those species, even those not statistically differential, with the results showing the most modulated ones with respect to the control healthy skin, to verify any interesting variation trend between treated and untreated skin. The ten most under- and ten most over-modulated OTUs were identified for both T0 and T1 groups. For those OTUs, the results were among the most modulated only for one of the two groups, the logFC variation was also verified for the other group and compared. 

### 4.7. Shotgun Sequence Analysis 

Since the heterogeneity of the body areas where the microbiota was sampled could influence the significance of the results, a multi-step data analysis was conducted for the Shotgun sequences. At first, the microbiotas of T.0 and T.1 groups were compared with each other, regardless of the body area sampled, to find possible variations common to all the sampling sites. Then, the comparison between T.0 and T.1 microbiotas was performed concerning the body district from which the samples were collected to find any interesting site-dependent variations. 

Regardless of the body area, the microbiota before treatment (T.0) was compared with the microbiota after treatment (T.1), as described below.

The alpha and beta diversity indexes were evaluated by Gaya software using the raw OUTs table without any sample rarefaction.

For the alpha diversity evaluation, the observed OTUs and Chao1 indexes were used to estimate the richness within samples and the Shannon index for the evenness as well. The normality of index distributions was assessed through the Shapiro–Wilk test (α = 0.05). The normally distributed series were statistically compared using Welch’s *t*-test, whereas the not normally distributed series through the non-parametric Kolmogorov–Smirnov test, both with a significance threshold *p* ≤ 0.05. The alpha diversity indexes of T.0 and T.1 microbiotas were compared at all taxonomic levels. 

Similarly, the Bray–Curtis distances values, sensitive to the taxa abundances, were calculated to evaluate the diversity between samples, i.e., the beta diversity. Distances matrices and PCoA plots were produced for all taxonomic levels to highlight the clustering of microbiota samples. 

The abundances of all domains detected before and after treatment were statistically compared by performing the Welch’s *t*-test for normally distributed series, whereas the Kolmogorov–Smirnov test was used for not normally distributed ones (*p* ≤ 0.05). Deepening the analysis, the most prevalent taxa at upper (phylum and class) and lower (genus and species) taxonomic ranks were identified. Their abundances in T.0 and T.1 microbiotas were compared. The statistical significance of comparisons between the abundance distributions was evaluated through the paired Student’s *t*-test and Wilcoxon test (*p* ≤ 0.05 for both), respectively, for normal and non-normal distributions of percentages.

Finally, the DESeq2 differential analysis was conducted by the Gaia platform to identify the possible differentially represented OTUs between the treated, T.1, and untreated T.0 microbiotas.

On the contrary, concerning the body area sampled, because only a few samples were collected from each body district, the skin parameters were not registered and evaluated. Instead, the differential analysis DESeq2 was conducted for those body districts with enough swabs sampled. The differential OTUs were finally related at the evolutionary level elaborating the phylogenetic tree through the phyloT v2 software (phyloT database version 2022.3).

## 5. Conclusions

Regardless of the complexity of our study, the difficulties in comparing and integrating the results obtained from two different data analyses, and the diverse intrinsic characteristics of the two cohorts of patients, the following interesting findings can be highlighted both regarding the pathologic mechanisms and the efficacy of the LimpiAL treatment. The IP onset influences the skin microbiota principally by inducing the increase of some phyla, classes, and orders already part of the skin microbiota and, at the same time, the occurrence of new, probably pathology-dependent, families, genus, and species. LimpiAL 2.5%, as a unique treatment for the IP lesion for only 4 weeks, has evident beneficial effects on clinical skin conditions. The variation of the skin parameter scores, at least of the PASI score, is comparable to the one induced by the analogues of vitamin D, which is considered among the first-line therapies for IP. Therefore, we can infer that LimpiAL 2.5%, containing hyaluronic acid conjugated to a bacterial wall fragment (HAc-40), exerts the therapeutic effects of first-line therapies. The mechanism of LimpiAL action against the pathologic processes is related to its ability to modulate the skin microbiota. LimpiAL counteracts the stimulation induced by the pathology on some microbial taxa and, at the same time, increases the microbiota diversification, probably acting on molecular pathways common at many bacterial phyla such as the *Actinobacteria* and *Firmicutes* phyla. 

It strongly reduces some *Corynebacterum* species that, even not yet ascertained in literature, could have a crucial role also in the onset of the IP pathology. At the same time, the topical product application increases the diversity among the *Staphylococcus* genus, stimulating, among others, the increase of *S. pasteuri*. This germ, producing the bacteriocin known as pasteuricin, could exert a control action against the other members of *Staphylococcus* genus, especially *S. aureus* with a well-known pathological role in the inflammatory skin diseases. Therefore, thanks to its combined actions on the *Corynebacterium* and *Staphylococcal* determinants, LimpiAL 2.5% exerts an evident potential role in restoring the eubiosis condition of the IP affected skin and leads to the composition of the microbial community of the lesion sites nearest to the healthy microbial pattern.

Overall, all our results help to clarify the mechanisms of microbiota modulation induced both by the pathology and by the LimpiAL treatment and, at the same time, suggest the potential efficacy of the tested product as an alternative and valid treatment for inverse psoriasis lesions. 

Despite these promising results, the low number of patients recruited in this preliminary investigation and the monocentric character of the study prompt the necessity to scale up the investigation involving more sampling centers, patients, and samples to confirm our findings.

On the other hand, this initial trial was certainly necessary for the rational conceiving of a future well-dimensioned clinical study.

## Figures and Tables

**Figure 1 ijms-24-06339-f001:**
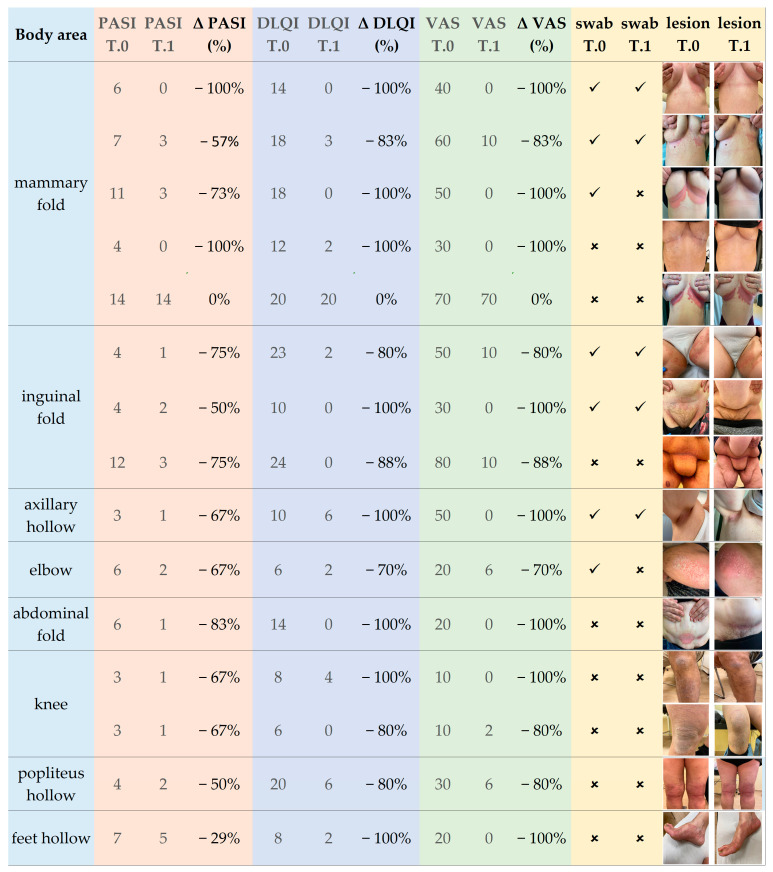
Clinical outcomes. In the first light-blue column, the body area sampled is indicated. The pink, blue, and green columns report the score and the percentage variation (Δ) of PASI, DLQI, and itch VAS indexes, respectively. In the yellow columns, the tick and cross symbols indicate whether the swab was collected or not collected.

**Figure 2 ijms-24-06339-f002:**
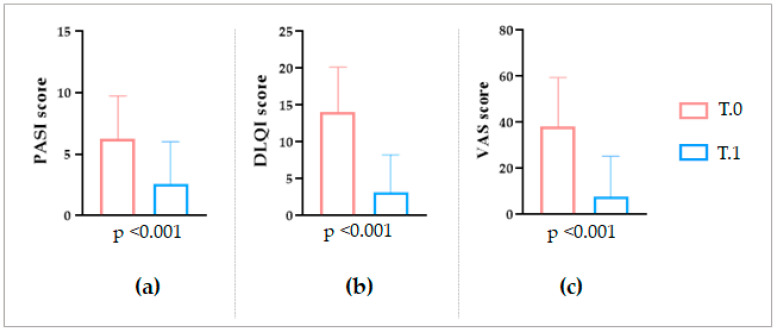
Statistical comparison between the (**a**) PASI, (**b**) DLQI, and (**c**) itch VAS scores registered before and after the LimpiAL treatment (Wilcoxon test, *p* ≤ 0.05).

**Figure 3 ijms-24-06339-f003:**
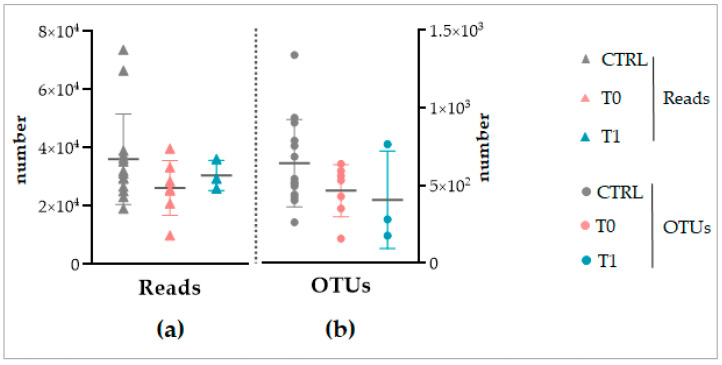
The sequencing output from the NGS Illumina MiSeq of 16S and ITS amplicons libraries. (**a**) The number of reads obtained. (**b**) The number of matched OTUs. The grey horizontal line highlights the average value.

**Figure 4 ijms-24-06339-f004:**
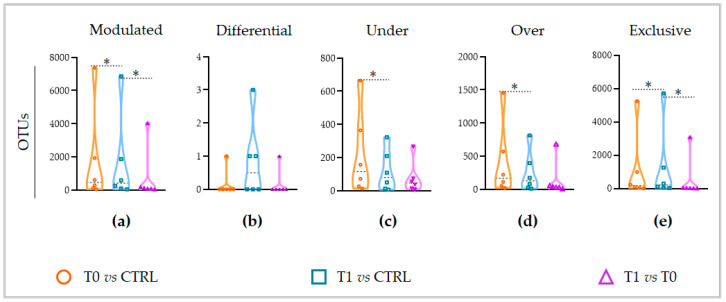
The modulation trends of OTUs in compared microbiota. The number of (**a**) modulated, (**b**) differential, (**c**) under and (**d**) over-represented, and (**e**) exclusive OTUs is reported with respect to each compared microbiota. The orange and blue colors indicate the OTUs of T0 and T1 microbiota compared to the CTRL ones. The violet color depicts the OTUs varied in the T1 microbiota with respect to the T0 microbiota. The asterisk indicates the statistical significance of comparisons, with * standing for *p* < 0.05.

**Figure 5 ijms-24-06339-f005:**
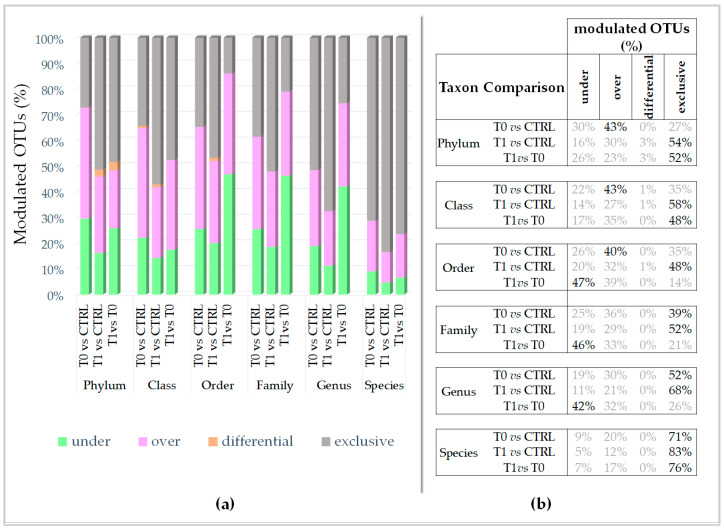
OTU modulation dependence of the taxonomical level. (**a**) Distribution of the under- and over-represented, differential, and exclusive OTUs along the taxonomical levels for all comparisons between microbiotas. (**b**) Abundance (%) of the under- and over-represented, differential, and exclusive OTUs is reported, and the highest abundance for each taxonomical level and comparison is highlighted in bold character.

**Figure 6 ijms-24-06339-f006:**
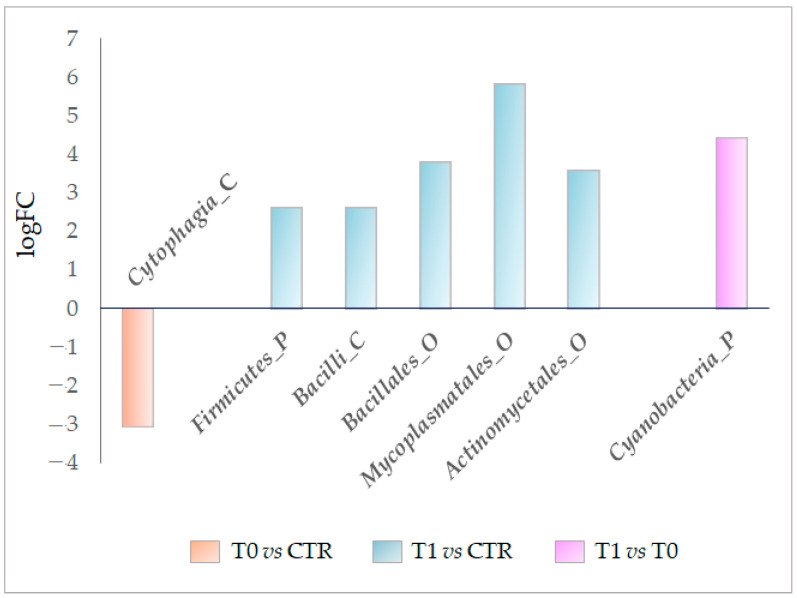
Taxa differentially modulated between the compared microbiotas (DESeq2 analysis, *p*-value, and FDR ≤ 0.05). After the underscore, the capital letter P, C, and O stand for phylum, class, and order, respectively.

**Figure 7 ijms-24-06339-f007:**
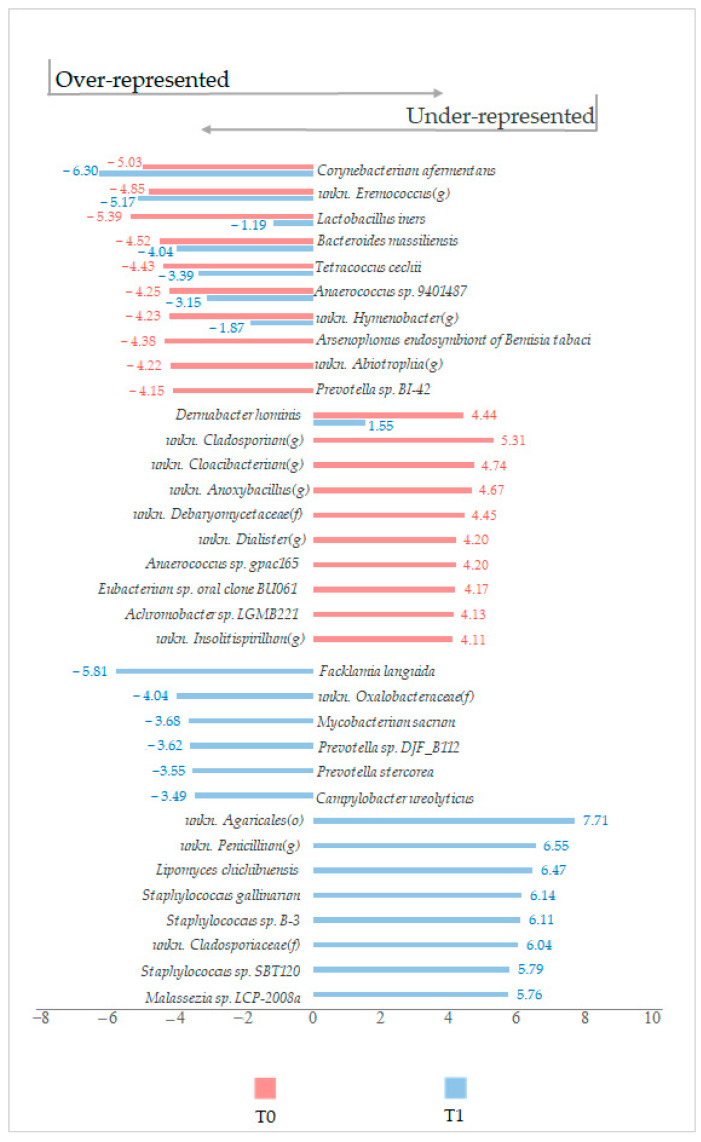
Comparison between the modulation trends detected in T0 and T1 microbiota for the ten most under (bars to the left side) and ten most over (bars to the right side) modulated OTUs.

**Figure 8 ijms-24-06339-f008:**
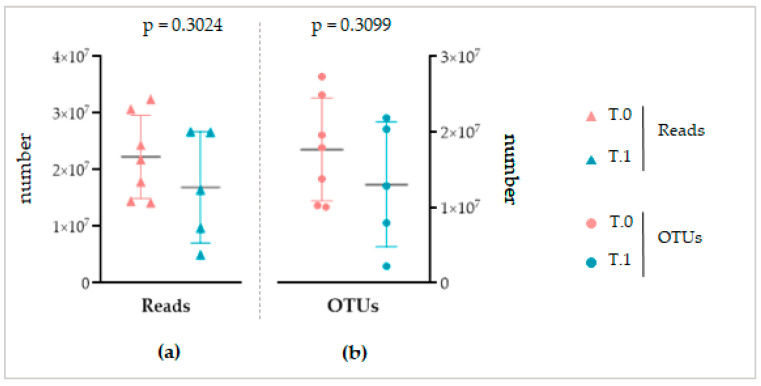
The sequencing output from the Shotgun sequencing. (**a**) The number of reads obtained. (**b**) The number of matched OTUs. The grey horizontal line highlights the average value.

**Figure 9 ijms-24-06339-f009:**
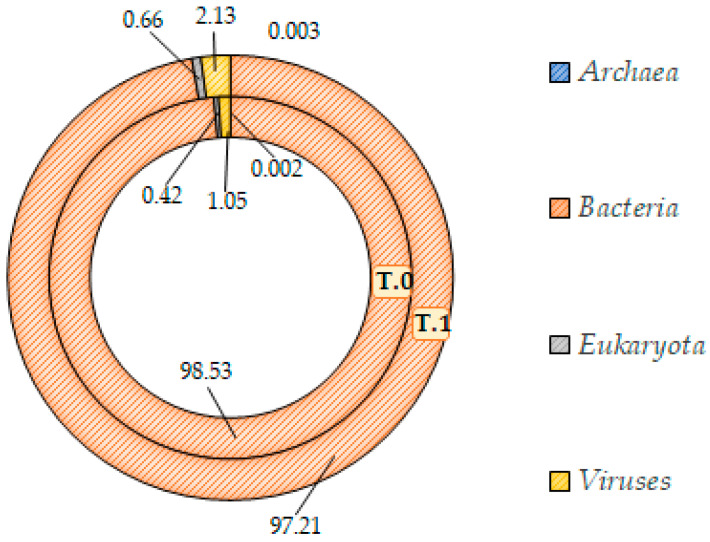
The percentage abundance of principal domains in T.0 (inner ring) and T.1 (outer ring) microbiota.

**Figure 10 ijms-24-06339-f010:**
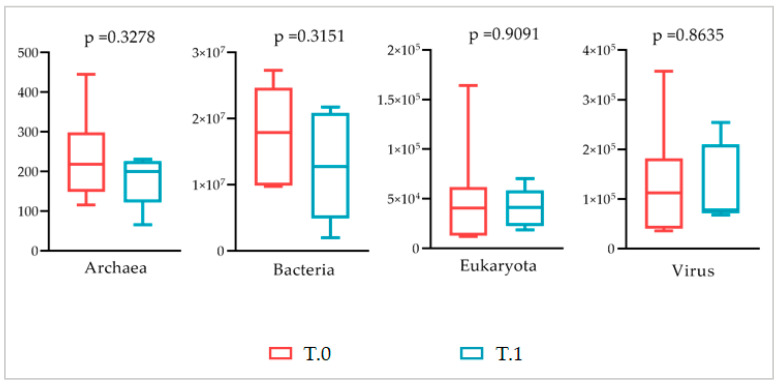
Statistical comparison between the prevalence of principal domains detected in the microbiotas before (T.0) and after (T.1) the LimpiAL treatment (unpaired Student’s *t*-test and Kolmogorov–Smirnov test, both with *p* ≤ 0.05).

**Figure 11 ijms-24-06339-f011:**
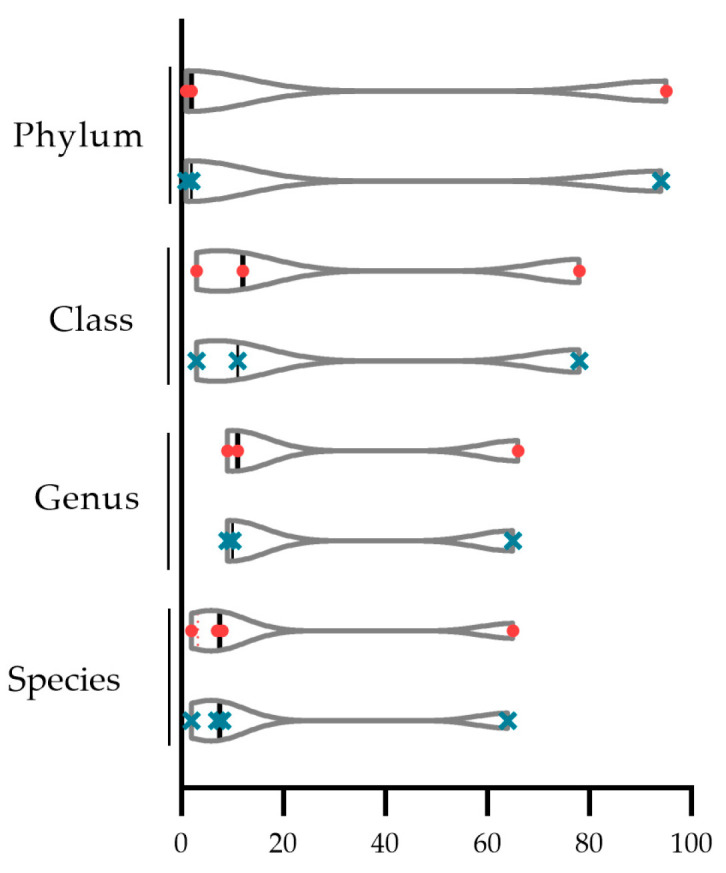
Statical comparison between the percentage distributions at the upper (phylum and class) and lower (genus and species) taxonomic levels (Student’s *t*- and Wilcoxon test with *p* ≤ 0.05). Red and blue symbols depict respectively T.0 and T.1 taxa distributions.

**Figure 12 ijms-24-06339-f012:**
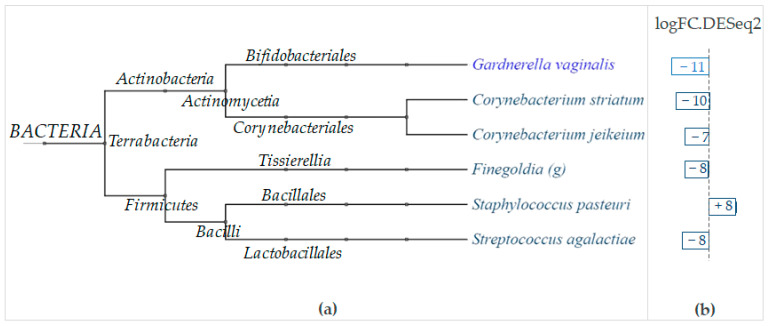
Taxa differentially modulated by the LimpiAL treatment in the microbiota of inguinal (light blue) and mammary fold (dark blue). (**a**) The phylogenetic relationship between modulated microbes (**a**) and the type and level of modulation induced (logFC) (**b**) are reported. Bars oriented to the left with negative numbers indicate a significant decrease of the detected taxon; bars oriented to the right with positive numbers highlight the increment.

**Figure 13 ijms-24-06339-f013:**
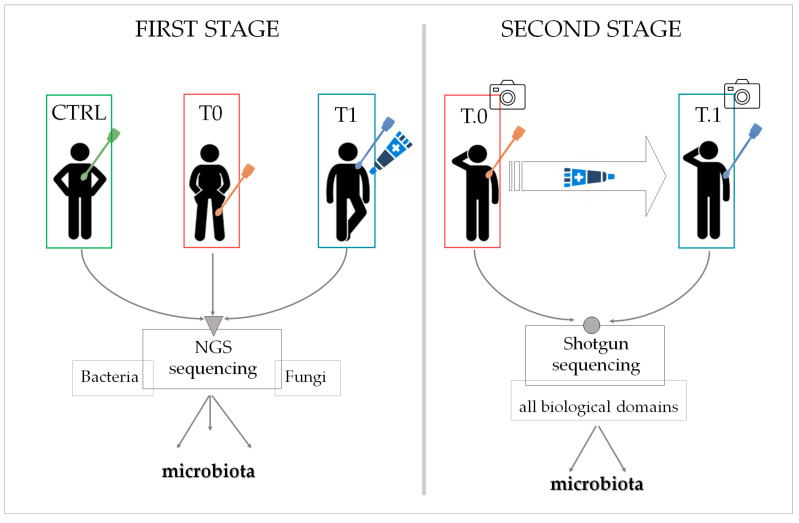
Study set-up and workflow. The different skin conditions and treatments are highlighted by colors: green for patients with healthy skin (CTRL), orange for IP patients, and light blue for IP patients treated with LimpiAL 2.5%. The same colors are used for related swabs. The application of LimpiAL 2.5% is graphically represented by the medical device blue tube. The camera icon indicates that photos of sampled areas have been collected. The identical patient’s icon is used when the same patients are considered in different groups.

**Table 1 ijms-24-06339-t001:** Statistical comparison between the alpha diversity indexes of T.0 and T.1 microbiotas at all taxonomic levels (Welch’s *t*-test and Kolmogorov–Smirnov test, both with *p* ≤ 0.05).

	T.0 vs. T.1						
	Domain	Phylum	Class	Order	Family	Genus	Species
OTUs number	>0.99	0.06	0.31	0.55	0.48	0.40	0.30
Chao1	>0.99	0.09	0.08	0.24	0.84	0.38	0.25
Shannon	0.91	0.75	0.74	0.70	0.69	0.67	0.69

**Table 2 ijms-24-06339-t002:** Comparison between the percentage abundances of more represented taxa in T.0 and T.1 microbiota.

	Taxon	T.0 (% ± SD)	T.1 (% ± SD)
Phylum	*Proteobacteria*	95 ± 3.6	94 ± 3.8
	*Actinobacteria*	2 ± 1.8	1 ± 1.0
	*unkn, Viruses(d)*	1 ± 1.2	2 ± 3.0
Class	*Betaproteobacteria*	78 ± 3.0	78 ± 2.7
	*Gammaproteobacteria*	12 ± 0.9	11 ± 1.0
	*Alphaproteobacteria*	3 ± 0.9	3 ± 0.6
Genus	*Cupriavidus*	66 ± 3.0	65 ± 4.2
	*Ralstonia*	9 ± 0.9	9 ± 1.1
	*Pseudomonas*	11 ± 0.2	10 ± 0.6
Species	*Cupriavidus metallidurans*	65 ± 2.9	64 ± 4.1
	*Ralstonia pickettii*	7 ± 0.4	7 ± 0.4
	*Pseudomonas aeruginosa*	8 ± 0.6	8 ± 0.7
	*Pseudomonas fluorescens*	2 ± 0.2	2 ± 0.3

## Data Availability

The complete analysis dataset is available by requesting directly to the corresponding author.

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
