# Peer review of "Efficacy and Microbiota Modulation Induced by LimpiAL 2.5%, a New Medical Device for the Inverse Psoriasis Treatment"

_ijms, 2023, doi:10.3390/ijms24076339_

Round 1

Reviewer 1 Report

The study is interesting, but the sample size is too small to have adequate results. It is necessary to increase the number of subjects to improve the study, and please provide the formula by which the sample size was calculated.

Author Response

We agree with the reviewer about the low number of patients and samples included in our study. Unfortunately, as indicated in the section Material and Methods, this study was conducted in the middle of the COVID-19 pandemic period between February 2021 and January 2022 and a lot of sanitary restrictions played against the enrollment procedures. Despite the surely major number of patients we initially calculated using the Software GPower v. 3.1.9.4, then our target was actually impossible to reach. Only one center agreed to conduct samplings, the turnout of patients specifically for the dermatological care was very low and many patients did not return after the first visit, often due to the covid and long infections.

However, we decided to publish in any case our research also on the basis of the fact that other comparable studies regarding psoriasis were already in literature published with similar low numbers of recruited patients (doi:10.2147/CCID.S374871, doi:10.1038/s41522-017-0022-5, https://doi.org/10.1186/s40168-018-0533-1, DOI: 10.1016/j.biopha.2021.112327).

In any case we amended our manuscript  better highlighting in the "Conclusions" section the preliminary nature of our research and the necessity to implement it to confirm the results especially from the statistical point of view (lines 845-854).

Reviewer 2 Report

Introduction is too long. Since this is a research article, introduction should be shortened by 40% at least. Just saying the difference between IP and normal psoriatic lesions and the explaining LimpiAL 2.5% are good enough

The information of LimpiAL is lacking. Its structure and biological, immunological functions should be explainedin more detail in Introduction and Discussion.

How the decrease of Corynebacterium increase of Staphylococcus by LimpiALmay contribute to the improvement of IP should be explained in more detail.

In introduction and discussion, the authors should describe how they manipulate the resuls in the alteration of cutaneous microbiota by LimpiAL ito apply for the treatment of IP.

Author Response

Introduction is too long. Since this is a research article, introduction should be shortened by 40% at least. Just saying the difference between IP and normal psoriatic lesions and the explaining LimpiAL 2.5% are good enough

The introduction section was shortened as suggested.

The information of LimpiAL is lacking. Its structure and biological, immunological functions should be explainedin more detail in Introduction and Discussion.

LimpiAL 2.5% formulation was more detailed both in the Introduction (lanes 77-95) and Discussion sections (lanes 443-454). In addition, all the LimpiAL ingredients were listed in Table S1.

How the decrease of Corynebacterium increase of Staphylococcus by LimpiAL may contribute to the improvement of IP should be explained in more detail.

We added the required informations into  the Conclusions section at lanes 831-844

In introduction and discussion, the authors should describe how they manipulate the resuls in the alteration of cutaneous microbiota by LimpiAL ito apply for the treatment of IP.

We amended the introduction (lines 96-114) and discussion section (lines 493-504) as suggested.

Round 2

Reviewer 1 Report

With the clarification made, the article can be published.

Reviewer 2 Report

Nothing to do